# Abusive Head Trauma Animal Models: Focus on Biomarkers

**DOI:** 10.3390/ijms24054463

**Published:** 2023-02-24

**Authors:** Rahul M. Nikam, Heidi H. Kecskemethy, Vinay V. R. Kandula, Lauren W. Averill, Sigrid A. Langhans, Xuyi Yue

**Affiliations:** 1Diagnostic & Research PET/MR Center, Nemours Children’s Health, Wilmington, DE 19803, USA; 2Department of Radiology, Nemours Children’s Health, Wilmington, DE 19803, USA; 3Nemours Biomedical Research, Nemours Children’s Health, Wilmington, DE 19803, USA

**Keywords:** abusive head trauma, animal model, biomarker, neurodegeneration, reactive oxygen, *N*-methyl-D-aspartate receptor, glia

## Abstract

Abusive head trauma (AHT) is a serious traumatic brain injury and the leading cause of death in children younger than 2 years. The development of experimental animal models to simulate clinical AHT cases is challenging. Several animal models have been designed to mimic the pathophysiological and behavioral changes in pediatric AHT, ranging from lissencephalic rodents to gyrencephalic piglets, lambs, and non-human primates. These models can provide helpful information for AHT, but many studies utilizing them lack consistent and rigorous characterization of brain changes and have low reproducibility of the inflicted trauma. Clinical translatability of animal models is also limited due to significant structural differences between developing infant human brains and the brains of animals, and an insufficient ability to mimic the effects of long-term degenerative diseases and to model how secondary injuries impact the development of the brain in children. Nevertheless, animal models can provide clues on biochemical effectors that mediate secondary brain injury after AHT including neuroinflammation, excitotoxicity, reactive oxygen toxicity, axonal damage, and neuronal death. They also allow for investigation of the interdependency of injured neurons and analysis of the cell types involved in neuronal degeneration and malfunction. This review first focuses on the clinical challenges in diagnosing AHT and describes various biomarkers in clinical AHT cases. Then typical preclinical biomarkers such as microglia and astrocytes, reactive oxygen species, and activated *N*-methyl-D-aspartate receptors in AHT are described, and the value and limitations of animal models in preclinical drug discovery for AHT are discussed.

## 1. Introduction

Abusive head trauma (AHT), also called non-accidental head injury or shaken baby syndrome, is a form of child abuse where a perpetrator violently applies repeated acceleration–deceleration forces to an infant with or without head blunt impact. AHT is the leading cause of death from trauma in children under the age of 2 years [1,2]. The median age of AHT victims is 4 months [3]. AHT has a high mortality rate of around 25% and morbidity incidence of 50% in survivors [4,5]. Subdural hematoma (SDH), cerebral ischemia, retinal hemorrhage, and skull fractures are the most common pathologic consequences of AHT [6]. In the United States, 13–36% of AHT victims die from injuries. In addition, most survivors suffer permanent physical, neurological, and mental disabilities including cerebral palsy, epilepsy, depression, anxiety, and posttraumatic stress disorder [7,8]. AHT in ages 0 to 4 years has been estimated to add USD 13.5 billion in societal costs each year [9].

Early and accurate diagnosis is critical, but correctly diagnosing AHT is challenging clinically and radiologically, even for experienced and astute physicians. Clinical symptoms may be subtle and histological data for the patients are often lacking. Jenny et al. [10] summarized the inflicted traumatic brain injury (iTBI) cases of 173 children younger than 3 years old and showed that physicians missed 31% of AHT cases on initial presentation. Among the missed patients, around one-third were injured again before the iTBI was confirmed, and 41% of the missed cases showed medical complications related to iTBI. Notably, four of the five deaths may have been preventable had the diagnosis been timely. Some studies also reported that approximately half of the children with iTBI did not present with any external trauma [11,12], while others showed only subtle indications of trauma. Conversely, an incorrect diagnosis may have significant implications on social and familial outcomes, such as infants being removed from their homes and parents losing child custody [10,13,14]. The missed and inaccurate diagnosis of AHT places the child at risk due to possible ongoing abuse and potentially life-threatening outcomes [10,15]. A particular diagnostic challenge is that abused children are usually too young to provide an adequate history to explain their symptoms [13,16]. Perpetrators are either unaware of the harmful behavior or unlikely to provide truthful confessions of trauma. Infants often show neither external signs of injury nor present a history of trauma, with non-specific symptoms such as vomiting and fussiness and a normal physical exam [11,12,17].

The diagnosis of AHT is a medical diagnosis formulated by a multidisciplinary collaborative effort considering all facts and evidence. It signifies that accidental and disease processes cannot plausibly explain the etiology of a child’s injuries. As aptly mentioned in the consensus statement on abusive head trauma in infants and young children by Choudhary et al., “A diagnosis of AHT is a medical conclusion, not a legal determination of the intent of the perpetrator or, in the false hyperbole of the courtroom and sensationalistic media, ‘a diagnosis of murder’” [18].

Imaging approaches pose additional diagnostic challenges because current imaging modalities do not provide independently specific or diagnostic results for AHT [18]. Computed tomography (CT) is the examination of choice in the initial evaluation of pediatric head trauma. However, early cranial CT in the setting of suspected AHT cases lacks sensitivity in detecting petechial hemorrhages, non-hemorrhagic strain, shear injury, ischemic edema, and ligamentous injuries of the craniocervical junction [19]. Conventional magnetic resonance imaging (MRI) is relatively less sensitive to subarachnoid hemorrhage and fractures and has a lower sensitivity for acute hemorrhage than CT. Sometimes, AHT does not show any visible findings on CT or MRI [2,20,21] since AHT cases are often asymptomatic. A missed diagnosis may lead to a catastrophic consequence; specifically, children younger than two years of age have a high mortality rate from AHT. In this review, we summarize the challenges physicians face in diagnosing AHT, evaluate the need for improved biomarker discovery, and discuss the potential of animal models in improving our understanding of molecular mechanisms mediating brain injury in AHT and preclinical drug discovery for AHT.

## 2. Clinical Challenges

Approximately 35 per 100,000 children younger than 1 years old are subject to AHT every year, and nearly 25% of children with AHT die [22,23]. It is challenging to diagnose AHT in terms of both social responsibility and medical accuracy. From the victim side, very young children (typically younger than 2 years old) are not able to provide verbal information about what happened to them. As a result, the caregiver may give inaccurate information and even fabricate a misleading history of the victim. In addition, clinicians may be biased in determining whether abuse occurred in a particular scenario. Failure to diagnose AHT puts a child at risk. Inaccurate conclusions, on the contrary, may wrongly remove a child from the custody of parents or guardians. In clinical practice, AHT cases usually present a spectrum of signs and symptoms, including such non-specific signs as vomiting and fussiness. All these scenarios make a timely and accurate diagnosis of AHT especially challenging.

The externally validated clinical prediction rules, such as Predicting Abusive Head Trauma and the Pittsburgh Infant Brain Injury Score, which can help avoid unnecessary testing and misdiagnosis, are usually used for early recognition of AHT. These clinical prediction rules facilitate estimating AHT probability when screening high-risk children without a trauma history [24]. Surveying a complaint-directed history and physical exam are the critical initial steps in identifying AHT cases. Clinical signs that support a diagnosis of AHT include rib, long bone, and skull fractures. Most AHT victims show retinal and subdural hemorrhages [18]. Non-contrast head CT is considered the first imaging choice for unexplained brain injury due to its short scan time without sedation and rapid determination of the necessity of neurosurgical intervention. MRI is sensitive in detecting diffuse axonal damage and delineating ischemia, parenchyma injuries, and cerebral edema, which are common in AHT [24]. Limitations of MRI are that the child is not allowed to take any food several hours before an MRI scan, and sedation is usually required for pediatric patients, which raises concerns for the developing brain [25,26]. Compared with CT, MRI may help differentiate subdural hemorrhage from benign subarachnoid space injury. Disproportionately large heads supported by relatively weak necks in children make cervical injuries common in suspected AHT cases; therefore, spinal imaging of soft tissues with MRI is recommended to support the diagnosis of AHT [24]. In comparison, cranial ultrasonography that neither involves radiation exposure nor requires sedation lacks sensitivity and specificity in diagnosing suspected AHT [22,27,28]. Clinicians must evaluate the patient history, conduct a differential diagnosis, and perform an evidence-based, comprehensive analysis of all factors to minimize misdiagnosis. At the same time, no single imaging modality can precisely diagnose all AHT [18,29,30,31].

## 3. Clinical Biomarkers of AHT

Due to the challenges of the clinical diagnosis of AHT, tremendous effort has been made to identify biomarkers that can aid in its clinical diagnosis. In 1998, Shannon et al. reported autopsy findings of 14 children (aged 1–27 months) who died from shaken baby syndrome [32]. Results showed *β*-amyloid precursor protein (*β*-APP)-positive axons in the cerebral white matter of all AHT cases, suggesting axonal injury. Cervical spinal cord and nerve injury with *β*-APP-positive axons were also present in most AHT cases, indicating that extension–flexion injury to the spinal cord may be critical in the pathogenesis of AHT. Dolinak and Reichard examined brains with an inflicted head injury in infants and young children for *β*-APP [33]. They found that *β*-APP immunohistochemistry was much more sensitive in detecting injured axons than hematoxylin and eosin or silver staining. In another short-surviving head injury study, even subtle morphologic axonal injury changes were detected as early as 2–3 h after injury by *β*-APP immunostaining [34]. Notably, while iTBI is sensitive to *β*-APP immunostaining to reflect axonal injury, other mechanisms causing axonal injury can also lead to positive *β*-APP immunostaining. Careful interpretation of *β*-APP immunoreactivity is critical since normal structures such as glia, dorsal root ganglion cells, and leptomeninges can also be positive. In addition, other factors such as global hypoxic-ischemic injury and children who survive resuscitation to ventilator support may also show extensive axonal staining. Therefore, *β*-APP staining should be interpreted carefully as a biomarker of AHT [35]. Since the *β*-APP immunostaining starts to fade about 7 to 10 days after injury, the authors proposed alternative biomarkers, such as the presence of macrophages and reactive astrocytes, to identify the injuries by staining for the cluster of differentiation 68 (CD68) and glial fibrillary acidic protein, respectively. In a study by Satchell et al., the concentration of cytochrome c, an electron transport chain component, was measured in 167 cerebrospinal fluid (CSF) samples of 67 children over 0–10 days after traumatic brain injury (TBI); among these, 15 patients were diagnosed with child abuse [36]. Results showed that increased CSF cytochrome c was independently associated with iTBI, indicating that the neuronal apoptosis associated with cytochrome c release is a prominent feature in child abuse cases. Monitoring CSF cytochrome c may be used to evaluate treatment therapy. In a similar study involving 37 patients with TBI (seven were diagnosed as AHT cases) admitted to the intensive care unit, the CSF level of cytochrome c was measured at four intervals (0–24 h, 25–48 h, 49–72 h, and >72 h after injury) by enzyme-linked immunosorbent assay [37]. Results showed that peak cytochrome c levels peaked at 49–72 h and were independently correlated with AHT. Increased cytochrome c levels in CSF predicted poor outcomes after TBI in pediatric populations, which indicated that apoptosis might play an important role in this particular population of pediatric brain injury [37]. Furthermore, the study found that the peak CSF cytochrome c level was significantly higher in AHT patients compared to accidental TBI cases. Neurotoxin biomarkers, including glutamate and quinolinic acid in CSF, have been reported in inflicted pediatric brain injury. The concentrations of neurotoxins in CSF of the inflicted brain injury were higher than those in non-inflicted TBI, yet neuroprotectant levels were less increased compared to non-inflicted TBI [38,39].

Newell et al. reported macrophage and lymphocyte activation in CSF after TBI [40]. This retrospective study included 66 patients with severe TBI; 17 were AHT cases (1 month–16 years old). CSF levels of macrophage activation marker-soluble CD163 (sCD163), iron deficiency marker ferritin, and soluble form of interleukin-2 receptor *α* (sIL-2R*α*) were measured by enzyme-linked immunosorbent assay at two points (17 h and 72 h). Results showed that markers of macrophage/microglia activation (sCD163 and ferritin) increased following pediatric TBI. CSF ferritin was higher during the first time point assessed, while sCD163 was higher during the second time point. No difference was observed in CSF sIL-2R*α* levels between TBI patients and the control group at the two points; however, the sIL-2R*α* levels in the CSF were highly correlated with sCD163 and ferritin levels. Similar studies have shown that activated microglia [41] or oligodendrocytes [42] released ferritin early, followed by an sCD163 increase. These studies also found that young pediatric patients, including AHT patients and low Glasgow Coma Scale (GCS) cases, had high CSF ferritin levels, indicating that younger children and more severe injury cases had higher macrophage and microglial activation. A high CSF ferritin level was associated with poor outcomes at a young age, low GCS, and AHT cases. Su et al. reported on 27 TBI patients, including six AHT cases (7 weeks to 16 years old), and measured the CSF myelin basic protein (MBP) concentrations [43]. Results showed the overall MBP concentration of the TBI cases at 5 days postinjury (dpi) was significantly higher than the controls. The patients younger than 1 year had lower mean MBP concentrations than those older than 1 year due to a lower relative fraction of MBP in the immature brain of an infant. These findings imply that an MBP increase in infants may underestimate the injury severity compared with older pediatric patients or adults. The mean MBP concentrations in AHT patients were lower than in non-abusive TBI. However, in this study, the mean age of AHT cases was significantly lower than that of the non-abusive TBI, and brain maturity likely affects the MBP concentrations. The study demonstrated that axonal injury with increased MBP in pediatric TBI, including AHT, may represent a promising therapeutic target.

While biomarkers in CSF provide helpful pathophysiologic information on AHT, it is challenging to obtain CSF from pediatric patients during a clinical visit for silent brain injury. More accessible serum markers including neuron-specific enolase (NSE), astrocytic marker S100 calcium-binding protein B, and MBP are under investigation. The comprehensive information of the three makers may provide insight into the timing of the injury or raise awareness in physicians that silent brain injury may have occurred [44]. Furthermore, in 2018 the U.S. Food and Drug Administration approved the first blood test to aid in diagnosing mild traumatic brain injury (mTBI) in adults. The test works by measuring the levels of two brain-specific protein biomarkers, ubiquitin C-terminal hydrolase-L1 and glial fibrillary acidic protein. The two proteins are released from the brain into blood and measured within 12 h of mTBI [45]. However, in a pediatric TBI study involving 49 children (1 week–12.4 years old, 39 TBI children including 10 AHT cases, 10 controls), the subjects had blood collected for biomarker evaluation within 24 h of presentation. Results showed the ubiquitin C-terminal hydrolase-L1 concentrations were significantly different between the controls and severe and moderate TBI cases. A significant difference was not observed in the mTBI cases, indicating the complexity of applying adult mTBI biomarkers to pediatric populations [46]. Gao et al. used two-dimensional difference gel electrophoresis combined with mass spectrometry to compare the serum protein profile of 18 pediatric patients with mild AHT to 20 age-matched controls [47]. Results showed that serum amyloid A levels were significantly increased in the AHT cases. The study also compared serum amyloid A expression levels in children with mild AHT and moderate-to-severe AHT. There was no correlation between the serum amyloid A levels and injury severity. Serum amyloid A may serve as a biomarker to identify infants with mild AHT that might be missed by traditional CT or MRI diagnosis. In addition, serum amyloid A probably has a much longer half-life than the serum biomarker S100 calcium-binding protein B, which is less than 60 min.

NSE and MBP are the most promising biomarkers to screen for brain injury in well-appearing infants with AHT [48]. In addition to MBP, as mentioned earlier in CSF, MBP in serum has also served as a biomarker for pediatric brain injury. In a prospective case-control study including 98 well-appearing infants, Berger et al. reported that 14 patients were diagnosed with iTBI [49]. MBP has a specificity of 100% and sensitivity of 36% in identifying brain injury. MBP screening is expected to add additional value in evaluating AHT cases that might be missed at initial diagnosis [49]. In this study, the researchers found that it took a significantly longer time for the patients with iTBI to be sent to the hospital compared with caregivers of patients with no brain injury, highlighting the challenges in timely diagnosis of AHT with biomarkers and the increased risk of a repeat injury or death for children. A summary of typical biomarkers for AHT cases is outlined in Table 1.

## 4. Preclinical AHT Animal Models

Inflicted head injury by shaking trauma is an important research topic. Few research groups are involved in animal research to simulate human AHT cases [50,51]. Most studies use mechanical shaking methods to simulate AHT scenarios in mice, rats, piglets, and lambs, and most utilize constrained head movement with a single-plane rotation. This methodology does not mimic clinical AHT cases in which the perpetrators randomly shake the heads of babies in multiple directions. This makes translation of preclinical findings to pediatric assessment challenging. It is widely accepted that the large size of gyrencephalic brains and relatively weak cervical muscles may better reflect clinical AHT cases. As a result, many animal models focus on pathologic changes found in AHT. Here, we introduce typical AHT animal models and biomarker changes after injury.

### 4.1. Mice in AHT

Most preclinical studies use infant mice to study AHT. Rotational shaking is the primary mechanism to induce AHT in mice, while the shaking directions are variable. While not ideally imitating AHT in the clinic, the rodent models partially reflect the pathology in clinical cases such as symptomatic subdural and subarachnoid hemorrhage, SDH, cerebral ischemia, retinal hemorrhage, diffuse axonal injury, and neurological problems. Bonnier et al. reported an AHT model using 8-day-old mouse pups. The mouse pups were shaken for 15 s on a rotating shaker and sacrificed at different ages. The brains of the mouse pups were processed for histological analysis. Ex vivo analysis of the brain samples showed that at 31 days old, hemorrhage or cystic lesions of periventricular white matter, corpus callosum, brainstem, and cerebellar white matter were observed in 75% of the survivors. Hemorrhagic lesions were evident from postnatal day 13, while cysts developed gradually between days 15 and 31. Reactive astrogliosis and microgliosis, an indication of neuroinflammation, were observed in the focal destructive white matter lesions. The study found that pretreatment of the shaken mouse pups with the *N*-methyl-D-aspartate receptor (NMDA) receptor antagonist MK801 alleviated the white matter damage, suggesting NMDA receptor activation due to the excessive release of glutamate may play a role in the pathophysiology of the lesions [52]. Wang et al. reported rotational acceleration–deceleration TBI in developing mice [53]. The 12-day-old mice were subjected to 90° head extension–flexion sagittal shaking with an angular acceleration of 22,616.97 ± 3659.45 rad/s^2^ at 3 Hz frequency. Different repeats were used, including 30, 60, 80, and 100. Results showed that the repeats and severity of injury significantly impacted the mortality rate and return of the righting reflex. At 30 rotational acceleration–deceleration injuries (RADi), no mouse pups died; at 60 RADi of repeated head shaking, the mouse pups developed apnea and bradycardia immediately. A decreased survival rate was observed at a higher shaking speed. The expression levels of both astrocytes and microglia were significantly increased at 3 dpi, particularly in the ventral pons. These results demonstrated an endogenous pro-inflammatory response and glial activation after acceleration–deceleration injury. Neuronal degeneration by silver staining was observed in the cerebral cortex and olfactory tubercles at 30 dpi following an RADi of 60 pounds per square inch by 60 exposures. This rotational head acceleration–deceleration injury model in neonatal mice partially mimicked the pathophysiological and behavioral changes in pediatric AHT and provided a good model for long-term study of the secondary rotational acceleration–deceleration-induced brain injury in developing animals. Cerebral blood perfusion (CBP) was significantly reduced and had not fully recovered until 24 h. A severe reduction in CBP implies that secondary brain damage caused by ischemia/hypoxia exists. The sudden sagittal acceleration–deceleration rotational movement induced shear stress to damage superficial vessels, SDH, subarachnoid hemorrhage (SAH), and ventral brain injury. Reports showed that shear stress and hypoxia could change tight junction proteins of the endothelium, which leads to cerebral edema through BBB disruption and fluid extravasation [54,55]. In addition to pro-inflammatory changes and microglial activation in this rotational acceleration–deceleration animal models, other clinical AHT cases and animal models of brain injury reported an inflammatory response and diffuse gliosis [52,56]. Diffuse axonal injury (DAI) characterized by axonal swelling and varicosities has been reported in clinical cases with AHT [32,56,57,58]. In the reported rotational acceleration–deceleration mouse pup model, DAI was seldom observed 30 dpi; however, progressive neuronal degeneration in the cortex and olfactory tubercles was present. The researchers suggested that a long-term study may be required to confirm DAI because the injured axons in the developing brains exhibit a graded response to injury severity [59]. The neuronal degeneration probably affects long-term neurological and behavioral function since the report showed some neurobehavioral deficits in adulthood following TBI in pediatrics [60]. Kane et al. reported an impact acceleration mice model with mTBI under light anesthesia. No scalp incision and protective skull helmets were involved. In this animal model, skull fractures and intracranial bleeding are rare without evidence of seizure and paralysis. However, mild astrocytic activity and increased phospho-tau levels were observed with BBB disruption [61].

The brain structures of mice and humans are very different. The mouse AHT models are not ideal for simulating clinically obtained AHT. Unlike humans, mouse pups do not exhibit pericerebral bleeding. Correlating the time lag in mice between the shaking and the development of bleeding and atrophy with clinical findings is still poorly understood. Mouse AHT biomarkers center on reactive astrogliosis, microgliosis, and the NMDA receptor. Despite these limitations, mouse models have utility: they are low cost, easy to maintain, and partially reflect the pathology in clinical cases such as symptomatic brain hemorrhage, ischemia, and neurological outcomes. These models also usually do not require a craniotomy, simplifying the experimental operation. Furthermore, genetically modified mice are readily available compared to large animals, and the use of these mice can improve our knowledge of testing novel hypotheses, elucidating pathological mechanisms, predicting long-term response, and identifying new therapeutic targets in AHT. Among the mouse AHT models, most studies lack consistent and rigorous characterization of shaking mechanisms. The model reported by Wang et al. better represents clinical AHT cases [53]. The mouse model uses a repetitive rotational head acceleration–deceleration mechanism with predefined parameters to partially mimic clinical AHT pathophysiology and behavior, including brain hemorrhage, hematoma, neuronal injuries, and cognitive impairment. Furthermore, inflammatory biomarkers were significantly elevated in this model compared with sham brains. One limitation is the study only uses sagittal shaking, while clinical AHT may occur within multiple planes around the body axis.

### 4.2. Rats in AHT

Bittigau et al. developed a model of head trauma in infant rats to study the mechanism of neurodegeneration in the developing brain. Two morphological types of brain damage were observed within 4 h and 6–24 h after trauma, respectively. This study showed that NMDA antagonists protected against primary excitotoxic damage but exacerbated the secondary apoptotic injury in 7-day-old rat pups. In the developing rat brain, apoptosis instead of excitotoxicity results in neuropathologic outcomes after head trauma. Radical scavengers and tumor necrosis factor inhibitors may help treat pediatric head trauma. Furthermore, the authors found that the severity of trauma-triggered apoptosis in the brains was age-dependent, and the immature brain was particularly vulnerable [62].

Huh et al. used the most common controlled cortical impact approach to mimic pediatric repetitive mild brain injury in the immature rats [63]. Postnatal day 11 rats were used to model AHT. A conventional controlled cortical impact tip or a customized rubber tip was used in the studies. In these models, the study was designed to avoid skull fractures. Axonal injury, neuroinflammation, and calpain activation were typically observed but seldom neuronal death. Repeated injury exacerbated the pathology as expected. Treatment with folic acid, minocycline, and FK506 was involved in several studies, but most showed limited efficacy compared with the adult TBI models in rodents [64,65].

Kawamata et al. used an experimental rat model of repeated mild shaking brain injury in rat pups to study neonatal cerebral microhemorrhages using susceptibility-weighted imaging and iron histochemistry. Results showed that postnatal day 7 rat pups had a significantly higher number of microhemorrhages than postnatal day 3 rat pups. In contrast, no microhemorrhages were detected in the control rat pups and pups 5 weeks after shaking. The staining pattern of iron-positive cells surrounding microhemorrhages lasted for a long time. Even the hemorrhagic signals disappeared, strongly suggesting focal hypoxic–ischemic insults. The open-field test showed that the shaken group had significantly lower numbers of line crossings and rearing events than those in the control group, indicating anxiety-related outcomes in adult rats [66]. Similar reports showed an excessive iron load increased anxiety-related behavior and caused brain injury via the formation of free radicals [67,68,69]. The strong iron-positive reaction probably indicated increased numbers of activated microglia and macrophages [70,71]. Recently, Daniel et al. used five-day-old Wistar rats to develop two AHT models [72]. The first model was subjected to low-intensity, high-duration rotating movements (one cycle per second, 15 min shaking per day for five consecutive days). The second model was subjected to high-intensity, low-duration anteroposterior movements (3.3 cycles per second with 10 periods of 6 s). The researcher compared the two models’ brain damage and biochemical marker changes. Results showed that hemorrhage was observed in 10% of the low-intensity, high-duration movements group, while this was much higher in the high-intensity, low-duration movements group (40%). The severity of brain damage is closely related to the magnitude of biochemical changes, including reactive oxygen or nitrogen species, oxidative stress, and energy metabolism.

In addition to AHT model development, some researchers used drugs to study the trauma mechanism at a molecular level. Smith et al. reported a shaking plus hypoxemia AHT animal model using postnatal day 6 rat pups [73]. Results showed that shaking of the rat pups led to cortical hemorrhages, cortical tissue damage, and the production of oxidative stress markers. An early study by the same group reported that the anti-excitotoxic glutamate release inhibitor riluzole alleviated cortical neurodegeneration, in contrast to the antioxidant tirilazad, which was ineffective [74]. Hanlon et al. reported the effect of minocycline, a broad-spectrum tetracycline antibiotic, on the treatment of repetitive TBI in 11-day-old Sprague-Dawley rat pups [75]. Results showed repeated injuries led to spatial learning and memory deficits and increased brain microglial and macrophage expression. Acute administration of minocycline in this AHT model decreased microglial/macrophage activation in the corpus callosum at 3 dpi, but this effect disappeared at 7 dpi. Interestingly, minocycline did not affect the traumatic axonal injury or axonal degeneration. In turn, this drug showed exacerbated injury-induced spatial memory deficits, while in adult brain-injured mice, minocycline treatment demonstrated efficacy in reducing impairments and injury-induced deficits [76,77,78,79]. In adult TBI and neonatal stroke animal models, minocycline treatment effectively induced lesion areas in the cortex [79,80,81,82]. The data suggest that in the repeated injury neonatal model of AHT, minocycline may not be an effective drug candidate in treating the acute period. However, the authors also pointed out that the dosing paradigm and detailed study of the effect of minocycline on microglia/macrophage polarization (pro-inflammatory vs. anti-inflammatory) phenotypes may underlie this interesting finding. As with mouse AHT models, rats are cost-effective, and researchers can easily use innovative techniques to create genetically modified strains to screen therapeutic targets in AHT. However, the notable differences in brain geometry, craniospinal angle, and white-to-grey matter ratio may lead to substantially different responses to AHT from subject to subject.

Overall, repetitive acceleration–deceleration forces are the most common cause of AHT. Daniel et al.’s low-intensity, high-duration rotating and high-intensity, low-duration shaking models in young rats clearly show morphological injuries and biochemical changes [72]. Furthermore, the severity of brain injuries is associated with the magnitude of the biomarker levels, which may provide some information on the relationship between the shaking forces, duration, and clinical outcomes.

### 4.3. Piglets in AHT

To better understand AHT, it is critical to use large animal models. The advantages of using large animal models, including monkey, lamb, and piglet models, are that these animals have a gyrencephalic brain supported by relatively vulnerable neck muscles, grey–white matter differentiation, and a physiological response similar to human infant brains. A 3- to 5-day-old piglet brain is roughly comparable to a 2- to 4-week-old infant in terms of activity, myelination, and growth. However, as with lambs, the piglet brains have an almost elliptical shape that is in line with the cervical spinal cord, significantly different from the rounder human brain forming a nearly 90° angle. Therefore, it is challenging to directly translate the studies to pediatric assessment due to the single direction rotation in most studies and different brain anatomy. Vester’s group recently reviewed animal models for shaking trauma and related findings on tissue damage. Their paper reviewed 12 articles published by two research groups involving lambs or piglets. Most animal studies only involved a single-plane rotational movement. Decreased axonal injury and death corresponded to increasing age and weight. The authors suggested that free movement in all directions simulating human infant shaking is required for future studies. In the review paper, the authors did not include shaking trauma animal models in rodents and claimed an inconclusive report of the methodology and result [83].

Friess et al. reported a moderate and non-impact rotational TBI model using 3- to 5-day-old piglets with multiple impacts at either 1 day or 1 week apart. Double rotation (average acceleration 55.2 and 54.3 krad/s^2^, respectively) by 1 day apart led to a significantly higher (43%) mortality rate compared with a single rotation (58.5 krad/s^2^) [84]. Meanwhile, the double rotation animals showed significantly longer unconsciousness duration than the control group on both day 0 and day 7. Retinal hemorrhage is one of the key features, and the surviving animals displayed behavioral deficits and axonal injury evaluated by *β*-APP staining [84]. Raghupathi and Margulies reported closed head injury in the neonatal pig of a 3- to 5-day-old model. The anesthetized piglets underwent rapid and inertial rotation (10–12 ms, single 110° axial rotation, average peak angular velocity of 250 ± 10 rad/s) of the head around the axial plane. Results showed five of the seven piglets were apneic without pupillary and pain reflexes immediately following injury. Severe coma was observed in all piglets, but they recovered by 6 h. SDH and SAH were evident in the frontal lobes, while limited intraparenchymal bleeding was present. Axonal injuries were observed in six of the seven studies of brain-injured piglets, which were mainly located in the central and peripheral white matter and middle brain. The study concluded that the immature piglet brain may be more vulnerable to traumatic axonal injury than the adult brain and therefore will have a higher mortality and morbidity rate. In addition to SDH, SAH, and traumatic axonal injury, the authors suggested that hypoxia may play a role in the distribution of traumatic axonal injury [85]. The researchers reported a follow-up study at different rotational speeds (mild level 142 rad/s and moderate level at 188 rad/s). Results showed behavioral deficits were observed in the moderate injury level during environmental exploration and visual-based problem solving. Furthermore, moderate injury levels led to axonal injury as determined by amyloid precursor protein immunohistochemistry [86]. Similar animal models were used to monitor cerebral blood flow, evaluate cerebral blood oxygenation [87], and assess treatment outcomes with folic acid [88]. The rotational direction and number of repetitions had significant consequences in the AHT cases. Coats et al. conducted ocular examinations in modest brain injury (shaking frequency of 2–3 Hz, the average peak-to-peak angular velocity of 22.71 ± 3.49 rad/s, and average peak angular acceleration of 606.21 ± 160.30 rad/s^2^) of infant piglets. Results showed ocular hemorrhages in 73% of the 51 piglets, of which about half were bilateral and primarily located near the vitreous area. Twenty-six cases with bilateral SDH showed ocular hemorrhages; only one had ocular hemorrhages in a unilateral SDH case. Ocular hemorrhages were accompanied by brain injury in all but two animals. However, the same group reported no ocular injury in a similar animal model, probably due to a much lower rotational velocity. Generally, all the above studies concluded that increasing force, duration, or repetition led to much greater frequency and more severe hemorrhages. Coronal shaking had less bleeding and axonal injury in frequency and severity compared with sagittal or transverse rotations [89,90].

While retinal hemorrhage is one of the principal findings in AHT, the exact cause of retinal hemorrhage from AHT is unknown. Umstead et al. hypothesized that retinal hemorrhages in AHT resulted from a combination of shaking forces and hypertension [91]. The team used eyes from young pigs to test the pressure required for sudden retinal hemorrhages. Subsequently, using either isolated shaking, hypertension, or combined conditions, the researchers created a computer model to simulate the loading. Results showed that hypertension or shaking alone did not generate adequate stress to induce retinal hemorrhages. Instead, combining the two forces without physical contact is a pivotal contributor to AHT.

AHT is often associated with posttraumatic disorders, including epilepsy, cognitive defect, and motor dysfunction [92]. Costine-Bartell et al. reported a synergistic, multifactorial injury cascades animal model in one-week-old piglets and one-month-old piglets to study the age-dependent role of seizures and edema in longitudinal tissue injuries after AHT [93]. The piglets at two developmental stages simulate clinical infant and toddler AHT cases. The multifactorial and brain volume scaled injuries, including cortical contusion, mass effect, subdural hematoma placement, kainic acid administration, brief apnea, and hypoventilation [94], were used to reflect the physiologic cases in children with severe AHT. Results showed that the outcomes and injury patterns were age-dependent. App-positive neurons were correlated with the hypoxic–ischemic-type damage but with different patterns: a higher amount of APP-positive neurons was observed in the ipsilateral hemisphere in the toddler piglets, while it was equivalent in the injured infant piglets. Infant piglets were clinically worse, with lower neurological scores than their toddler counterparts, while the seizure duration was not different among developmental stages. Furthermore, the study found that infant piglets underwent endogenous mechanisms to alleviate the bilateral injuries, while toddler piglets tended to limit the damage to a unilateral pattern. Combined with clinically relevant biomarkers, the model may bridge the gaps between injuries and therapeutic outcomes.

Several studies reported that the timeframe between two injuries and when to evaluate the impact postinjury might affect the trauma interpretation [84,89]. For example, more white matter injury and *β*-APP staining were detected for single rotated piglets surviving 5 days than those surviving 12 days [84]. At 6 h postinjury, no difference in the content of axonal injury was observed between episodic and continuous cyclic head rotations for 30 s [90]. However, at 24 h postinjury, the continuously rotated animals showed a significant increase in axonal injury. In the control group, no axonal injury was found. The axonal injury was found in all studies, although not always significantly differing compared with the control groups. Raghupathi and Margulies reported several shaking-related traumas and found no neuronal loss. There was no correlation between the velocity and density of axonal injury in the white matter tracts [85]. The same group reported that when the neonatal pigs were subjected to two consecutive rotations, the axonal injury was observed in the peripheral subcortical, central deep white matter of the parietal and temporal lobes, corpus callosum, hippocampus, and basal ganglia [59]. Specifically, more foci with multiple injured axons existed compared with a single rotation. In the moderately rotated piglets (average 62.9 krad/s^2^), Friess et al. found axonal injury in the olfactory tract, internal capsule, and germinal matrix. However, no axonal injury was observed in mildly rotated piglets (34.1 krad/s^2^) [86]. The same group reported that the most axonal injury was observed in the frontal lobes of injured animals with significantly higher *β*-APP levels in white matter [84]. Naim et al. had similar findings in a piglet injury model indicating the deep white matter of the frontal lobes, parietal and temporal lobes, or brainstem were the injured sites [88]. Eucker et al. found that the animal rotational direction significantly affected the outcomes: the transverse rotations resulted in more axonal injury than coronal or lower velocity horizontal rotations. More injuries were observed in both sagittal and transverse rotations than in coronal cycles. The latter showed minimal pathology. The axonal injury was more often observed in the anterior regions of the brain compared with other regional brain sections [95]. Coats et al. found that in cyclically rotated piglets, an injury occurred in 88.5% of the surviving animals. At 24 h postinjury, higher axonal injury was observed after continuous rotations for 10 s than 30 s. At the same time, compared with a single head rotation, the 30 s continuously rotated piglets had more hypoxic–ischemic injury [90]. Ibrahim et al. found that based on the mass scaled acceleration principle, there was a significant difference for both SAH scores and brain volumes of axonal injury when comparing the results of the 4-week-old piglets to published 5-day-old counterparts, while electroencephalogram responses between the two groups were similar [96]. Furthermore, higher rotational accelerations (61 krad/s^2^) resulted in more severe SAH, increased areas of ischemia, and more axonal injury compared with lower rotational acceleration (average 31.6 krad/s^2^) [96]. However, compared with real animal models of the two groups at similar acceleration rotations, the severity of SAH and axonal injury were similar. The authors concluded that the traditional mechanical engineering method of scaling by mass in the toddler does not apply to the developing infant brain.

Treatment studies in piglet AHT models have also been evaluated. Naim et al. studied the function of folic acid in a neonatal piglet model of TBI with 3- to 5-day-old female piglets [88]. The brain injury model was set up by rapid axial head rotation without impact. Two injured groups were involved in the study: one group received folic acid at a dose of 80 µg/kg by intraperitoneal injection 15 min postinjury, which lasted for six consecutive days, and the other injured group received an intraperitoneal injection of saline at the same time as the first injured group. Meanwhile, the study included two uninjured control groups: one group injected with folic acid and the other with saline, all following the same timeline as the injured animals. Results showed the injured group had significantly longer unconsciousness durations. Extensive neurobehavioral and cognitive testing including behavior, memory, learning, and problem solving were conducted on days 1 and 4 postinjury. The piglets were sacrificed on day 6 postinjury, and brain samples were processed for histological analysis. Results showed that seven of the 24 injured animals died due to palate fracture, cervical spine hematoma, pulmonary edema, or large SDH. For the animals monitored until day 6, the folic acid treatment group showed higher exploratory interest, better motor function, learning, and problem solving compared with the saline treatment piglets in the injured group on day 1 postinjury. However, functional improvements were not observed on day 4, which indicated that folic acid might increase early brain functional recovery in the non-impact head trauma model.

Limited piglet trauma models used multiple accelerations [90], which makes it less valuable for translating to human shaken babies because, in reality, many believe that the perpetrators shake the baby in a repeated and sudden acceleration–deceleration manner [97] instead of a single event. The head of the baby rotates in all directions although mainly in a sagittal plane along with possible chin–chest collisions [98]. While different shaken directions have been reported by several groups [89,90,95], none of the studies involved combining other rotation planes. The combination of various shaken planes at the same time may intensify the forces and deformations and exacerbate brain injuries. Therefore, most studies do not represent the main repeated back-and-forth movements in different directions and hardly translate to clinical diagnosis.

In summary, piglets have a gyrencephalic brain supported by relatively vulnerable neck muscles, similar to human infant brains. In addition, the piglet’s eyes have greater similarities to infant human eyes than most other animals. However, most studies used single-plane rotation instead of different rotation planes, making it challenging to translate the preclinical results to pediatric assessment. The piglet model reported by Coats et al. that used cyclic head rotations is similar to clinical AHT scenarios [90], although this model does not represent free movements in all directions. In addition, the model creates mild pathological and clinical AHT symptoms instead of the repetitive, sustainable injuries observed in severe AHT in the clinic. In pediatric AHT, it is widely accepted that repeated sudden deceleration in combination with acceleration causes intracranial injury. Therefore, there are insufficient data to predict the extent to which the piglet model results could be translated to clinical findings. Axonal injury evaluated by *β*-APP staining is used in both piglet models and clinical cases and may serve as a biomarker for AHT evaluation.

### 4.4. Lambs in AHT

Infant lambs were also used in the mTBI model. Anesthetized 7- to 10-day-old lambs were used to model AHT by physical shaking [98,99,100]. The animals were held under the axilla, then manually subjected to vigorous and multiple-episode shaking for 30 min. Multiple injuries, including BBB disruption, axonal damage, brainstem injury, and craniocervical junction damage, were observed during the studies. Finnie et al. reported similar young lambs subjected to a free shaking mechanism (10 × 30 s in 30 min) by humans [101,102]. Results showed manual shaking caused extra-axial hemorrhages in all lambs. Significant *β*-APP-positive neuronal perikaryons were observed in all injured lambs. However, several major injury indexes, such as total injury scores, hypoxic edema, and C-Fos immunoreactivity, were higher in the younger lambs than in the older ones. In a subsequent study, the authors found that retinal injury with increased glial fibrillary acidic protein expression, inner nuclear layer neuron injury, and increased *β*-APP levels were more widely seen in the younger lambs. The shaken lambs typically showed brain, spinal cord, and eye injuries. The lower-weight and younger lambs had a higher mortality rate.

For inflicted head injury by shaking trauma in shaken lambs, the animals were subjected to the acceleration–deceleration rotation without a direct impact on the head in any direction by adults, similar to the shaking of infants. Several studies used devices to measure the forces with a triaxial piezoresistive accelerometer and a motion-tracking sensor to monitor the accelerations [98,100]. The authors compared the animal models with 9-month-old infants based on their body weight. They concluded that trauma pathophysiology was comparable to young pediatric patients since both lamb and human infants have weak neck muscles, relatively large brains, and vast subarachnoid space, making brain movement within the skull relatively easy [97]. At the same time, both the infant and lamb brains have relatively higher water content and are not fully myelinated; therefore, the immature brain is more vulnerable to shearing injury [101,103]. However, the shape and orientation of the brains of lambs and human infants have significant differences: the lamb brain is in line with the cervical spinal cord and has a more elliptical shape; the human brain has a rounder shape and is almost vertical to the spinal cord. The difference may cause different trauma effects during shaking. One limitation is that the same injured lambs and control group were used for the three publications. Larger populations when reproducing the study may result in a more accurate interpretation of the results.

Overall, the free shaking mechanism applied by humans to the lambs in the study reported by Finnie et al. most closely resembled shaking in human babies. The lambs were held by adults with the head free for acceleration–deceleration rotation in any direction for a significant time (30 s) without a direct impact trauma. The studies partially answer the shaking outcomes in human conditions. Both human infants and lambs have weak neck muscles, a relatively larger brain compared with the entire body, and a higher brain water content, leading to likely shearing injury. The *β*-APP biomarker is also routinely used to evaluate clinical AHT cases, making the lamb model comparable to human infants. However, the lamb brain is more elliptically shaped and in line with the cervical spinal cord; in contrast, in humans, there is an almost 90° angle between the brain and the spinal cord. The brain shape and orientation of the two results in injury outcomes that are unclear. In addition, the small sample size and a lack of rigorous characterization of shaken kinematics make reproducibility a potential challenge and limit our ability to extrapolate these findings to human infants.

### 4.5. Other AHT Animal Models

Serbanescu et al. used an interesting natural shaking animal model to study retinal hemorrhages [104]. An 18 kg mixed-breed canine captured two 3- to 4-week-old kittens and one rabbit from a stray litter. The kittens and rabbit were bitten on the haunches and posterior spine, followed by four to six aggressive side-to-side shakes. The kittens and rabbit were killed by the canine and this was witnessed by one person. The deceased animals were subjected to pathological examinations after 1.5 weeks. Results indicated that the eyes of the two kittens and one rabbit did not show evidence of vitreous hemorrhage, retinal detachment, or retinoschisis. In addition, no retinal or optic nerve sheath hemorrhage was observed during the examination. The authors concluded that a more significant amount of force may be required for retinal hemorrhage due to their small eye size compared with a human infant. Another possible explanation is that the feline head and neck are better constructed to sustain acceleration–deceleration forces without injury [104]. Increased tau levels have been reported in pediatric TBI patients [105]. Alyenbaawi et al. set up an interesting TBI mode in a closed syringe with zebrafish larvae encoded with fluorescent tau protein. The team dropped a weight onto the plunger to mimic a shockwave-induced injury. Results showed the zebrafish larvae developed seizure symptoms, and the severity of seizures was correlated with the abnormal tau levels. Tau may serve as a potential biomarker for AHT, while further investigation in the AHT model with zebrafish larvae is required [106]. Eldridge et al. reported a focal impact TBI model in Xenopus laevis tadpoles with a pneumatic piston device. The model showed a secondary injury cascade, including neuroinflammation, oxidation, and BBB disruption [107]. A summary of typical biomarkers in preclinical animal models is outlined in Table 2.

## 5. Use of AHT Animal Models to Study Long-Term Outcomes in Adults

It is well known that an abusive environment in childhood is associated with individual anxiety behavior in adulthood. The biochemical changes in childhood of AHT animal models probably affect adults’ long-term neurological and behavioral functions. In a repeated mild shaking of the neonatal rat model, Kawamata et al. found that iron leakage surrounding microhemorrhages in the grey matter and iron-induced reactive oxygen species were observed, which caused long-term iron deposits and contributed to emotional abnormalities in adults and was an indication of anxiety-related behavior in adult rats. The animal model may shed light on the anxiety-prone state of AHT in adults [66]. In a subsequent study, the same group extensively studied how repeated shaking of neonatal rat pups affected long-term behavioral, hormonal, and neurochemical changes in adult rats [108]. The rat pups were shaken for 60 s, then rested for 60 s. The procedure was repeated five times. Half of the rat pups were shaken at postnatal days 3–7; the other half were shaken at postnatal days 8–14. The rat pups were housed until 8–10 weeks for further studies. Results showed transient microhemorrhages were observed in the grey matter of the hippocampus and medial prefrontal cortex. According to a similar animal model by the same group, leakage of free iron and iron-uptake cells surrounding microhemorrhages was presented [66]. Iron overload was reported to have long-term adverse effects [69,109], including increased superoxide production and mitochondrial dysfunction in the neonatal brain [110]. Behavioral tests showed the rat pups shaken on postnatal days 3–7 had significantly reduced locomotor activity and exploration behaviors than those shaken on postnatal days 8–14. Anxiety-like behavior was evident in the shaken group by the elevated plus maze (EPM) and the light/dark transition tests. Hormonal measurements in adult rats showed that the EPM induced significantly higher adrenocorticotrophic hormone and corticosterone responses. At the same time, the mineralocorticoid receptor expression level in the hippocampus was significantly reduced, which implied that downregulated mineralocorticoid receptors led to abnormal secretion of adrenocorticotrophic hormone and corticosterone. Neurochemical analyses showed the levels of dopamine, serotonin, 5-hydroxyindolacetic acid, and noradrenaline were increased in the dorsal part of the medial prefrontal cortex. This study clearly showed shaking of neonatal rat pups resulted in high anxiety-like behavior, an abnormal hormonal response, altered mineralocorticoid receptor messenger RNA expression, and monoamine in adulthood. Most recently, the same group assessed the change in sensitization in the anxiety–stress-related regions of adult rats by Fos immunohistochemistry after the neonatal rats were subjected to shaking brain injury. An EPM test was conducted to assess the psychological stress, including fear and anxiety, the rats naturally displayed. Results showed significantly increased Fos expression in the hypothalamus of the control and shaken groups when the rats were subjected to EPM, among which the shaken group had higher Fos expression than the control group. The results corroborated previous studies that the shaken rats exposed to the EPM had long-term hypersecretion of corticosterone and adrenocorticotrophic hormone in the serum of adults [108], which has been positively related to anxiety-like behaviors [111]. The study found that the psychological stressor EPM led to neuron activation in the ventral section of the bed nucleus of the stria terminalis. A positive correlation in Fos expression was observed between the ventral section of the bed nucleus of the stria terminalis and the parvocellular part in the shaken group, whereas the control group did not show such correlation. This study demonstrated that neonatal shaking brain injury caused persistent brain activity changes in adults when the rats were exposed to psychological stress. The data may provide meaningful information to study anxiety-prone states in shaken children.

## 6. Discussion

Given the high mortality and morbidity rate of AHT, robust and reliable translational investigations are clearly needed in pediatric abusive trauma. Using large animals to study AHT has been described by several research groups. The advantages of using large animals are their gyrencephalic brains mimicking the pediatric condition in terms of the amount of white matter, clinically relevant physiological monitoring, and pharmacological intervention. Limitations of rotational injury animal models include a lack of mechanical impact to obtain consistent and reproducible results, limited molecular tools in piglets and lambs, and less well-established behavioral outcome tasks than in rodent models. Most animal models selected for AHT are those likely to be appropriate to the specific neuropathology under investigation. However, large gaps still exist based on the currently utilized animal models. First, many studies lack consistent and reliable characterization of shaken mechanisms with low reproducibility or have difficulties with clinical translation. Second, the brain structures of rodents and large animals significantly differ from developing pediatric brains. Third, it is still unclear how secondary injuries impact the brain during the pediatric development period and long-term degenerative diseases. Several critical mechanisms in the secondary injury cascades were discussed [112]. The effect of the critical targets, including neuroinflammation, excitotoxicity, reactive oxygen species, axonal damage, and neuronal death on the different developmental stages in newborns, children, and adults, remain underinvestigated. A limited investigation was conducted on how genetic differences, including the basic sex variation, affect the outcomes. Another limitation is that most AHT models do not have a detailed description of the shaking (or rotational) angular velocity, acceleration, or if the injury was caused by random human force, which leads to reproducibility and rigor issues in preclinical studies. It is challenging for clinicians to translate and develop new therapies and extrapolate the animal study results to clinical AHT cases in a pediatric population for the following two major reasons: (1) the shaking mechanism in most animal models is not rigorously defined and (2) there are substantial structural differences between the developing infant brain and the animal brain. However, changes in biomarkers provide clues that the dysregulation may be related to brain injury in a more reliable and sometimes non-invasive way. To better understand the interdependency of the injured neurons and malfunction, an analysis of the cell types of the degenerated neurons and a study of long-term neurological outcomes during AHT in developmental brains are urgently needed. In addition to the currently reviewed key biomarkers in AHT (axonal injury, reactive oxygen species, activated NMDA receptor, microglia/astrocytes, hypoxic–ischemic), potential biomarkers detected in clinical AHT cases such as specific CSF concentration, S100, MBP, and cortisol level might be investigated to study the severity and predict the prognosis of AHT, serving as an alternative way of studying AHT. A positive screen may imply that the brain is injured and prompt the treating physician to perform further evaluation while confirmation of the diagnosis of AHT is being made. The reported rotational acceleration-deceleration models in animals may provide a good translational tool to identify biomarkers and evaluate therapeutic interventions related to pediatric AHT. Among the developed AHT animal models, the manual free shaking study reported by Coats et al. in lambs most closely resembled shaking in human infants [90]. The model allows maximal free movement of the head. Furthermore, the anatomical structures between infants and lambs are relatively similar: both have weak neck muscles, a proportionally larger brain compared with the entire body, and a higher brain water content, leading to likely shearing injuries and biomarker changes. *β*-APP is a favorable biomarker for axonal injury evaluation in AHT. Microglia and astrocytes are often assessed biomarkers to indicate brain inflammation during different stages in AHT models. The biomarkers may be helpful to prompt further screening for evidence of brain injury, although the markers are not specific to AHT. Improving the sample size in randomized animal studies, consistent and rigorous characterization of shaken mechanisms, longitudinal monitoring biomarker changes in serum and CSF, and validating the findings in the same species with different age ranges may improve the reproducibility of the model and translate the results to human infants. Furthermore, the biochemical changes in childhood of AHT probably affect adults’ long-term neurological and behavioral functions. Therefore, extensive neurobehavioral and cognitive testing in the favorable AHT models will greatly improve our understanding of the diagnosis, treatment, and management of clinical AHT.

## 7. Conclusions

AHT is a severe form of TBI in children. The correct diagnosis of AHT is challenging and requires a multidisciplinary approach. Current animal models show limitations in replicating clinical AHT reliably. Biomarkers after secondary injury in AHT may be helpful in screening tests, and for predicting outcomes and stratifying children with increased risks of clinical deterioration. Changes in biomarker levels may help identify AHT where it is challenging to assess brain injury by traditional physical exams or neuroimaging approaches. However, it should be noted that no biomarkers are specific to AHT. Spatially and temporally correlating the biomarker changes in animal models with human findings is still a massive challenge in studying AHT.

## Figures and Tables

**Table 1 ijms-24-04463-t001:** Biomarkers and findings in clinical AHT cases.

Authors	Number of AHT Cases/Total Cases	Age for All Cases	Findings	Biomarker	Time to Perform Biomarker Analysis
Shannon et al. [32]	14/14	0.5–27 months	The autopsy showed *β*-APP-positive axons were observed in the cerebral white matter of all AHT cases	CSF *β*-APP	Forensic autopsy records covered 15 years
Satchell et al. [36]	15/67	0.1–16 years	Increased CSF cytochrome c was independently associated with iTBI	CSF cytochrome c	0–10 days after injury
Au et al. [37]	7/49 *	5 weeks–16 years	Peak cytochrome c levels were independently correlated with AHT, and increased cytochrome c levels in CSF predicted poor outcomes after TBI in pediatric populations	CSF cytochrome c	CSF was collected daily for up to 7 days; samples were analyzed 0–24, 25–48, 49–72, >72 h after injury
Ruppel et al. [38]	4/37 **	0.1–16 years	Massive increases in CSF glutamate were found in children < 4 years old and child abuse victims Increased CSF glutamate and glycine were associated with poor outcomes Young age and child abuse were associated with high CSF glutamate concentrations after TBI	CSF glutamate	CSF samples were collected at the time of catheter insertion placement
Bell et al. [39]	3/17	2 months–16 years	AHT cases had increased admission CSF quinolinic acid concentrations compared with children with accidental mechanisms of injury	CSF quinolinic acid	On admission and 1–7 days after brain injury
Newell et al. [40]	17/83 ***	1 month–16 years	Markers of macrophage/microglia activation were increased Younger children and more severe injury cases had higher macrophage and microglial activation	Macrophage and microglia	CSF was collected every 12–24 h for up to 7 days after injury
Su et al. [43]	6/84 ****	8 days–16 years	Overall MBP concentration of the TBI cases at 5 days postinjury was significantly higher than the controls	CSF MBP	All patients’ GCS score ≤ 8 CSF was collected daily during the first 5 days
Gao et al. [47]	18/38 *****	Average 6.0 months	Serum amyloid A levels were significantly increased in the AHT cases, and serum amyloid A may serve as a biomarker to identify infants with mild AHT that might be missed by traditional CT or MRI diagnosis	Serum amyloid A	A mean of 9.2 h
Berger et al. [49]	14/98 ******	Less than 12 months	MBP has a specificity of 100% and sensitivity of 36% in the identification of the brain injury	Serum MBP	Not mentioned CSF was collected at the time of evaluation

AHT abusive head trauma, *β*-APP *β*-amyloid precursor protein, CSF cerebrospinal fluid, H&E hematoxylin and eosin, iTBI inflicted traumatic brain injury, MBP myelin basic protein, NSE neuron-specific enolase, TBI traumatic brain injury. * Includes 37 TBI patients and 12 controls. ** Includes 18 cases with severe TBI and 19 control cases. *** Includes 66 pediatric patients with severe TBI and 17 controls. **** Includes 27 TBI cases and 57 controls. ***** Includes 18 AHT cases and 20 controls. ****** Includes 14 AHT, 74 no brain injuries, 5 indeterminate patients, and 5 brain injuries not caused by AHT cases.

**Table 2 ijms-24-04463-t002:** Findings of preclinical AHT animal models and related biomarkers.

Author	Species	Age	Injury Mechanism	Findings	Biomarker
Bonnier et al. [52]	Mice	8 days	Shaken for 15 s on a rotating shaker	Hemorrhage or cystic lesions of periventricular white matter, corpus callosum, brainstem, and cerebellar white matter; pretreatment of the shaken mouse pups with the NMDA receptor antagonist MK801 alleviated the white matter damage	Reactive astrogliosis, microgliosis, NMDA receptor
Wang et al. [53]	Mice	12 days	A 90° head extension–flexion sagittal shaking with an applied angular acceleration of 22,616.97 ± 3659.45 rad/s^2^ at 3 Hz frequency, repeated 30, 60, 80, and 100 times	Acute oxygen desaturation and a severe reduction in cerebral blood perfusion, displayed reversible sensorimotor function, astrocytes and microglia were significantly increased at 3 days postinjury	Astrocytes and microglia
Bittigau et al. [62]	Rats	3–30 days	Weight drop injury	NMDA antagonists protected against primary excitotoxic damage but exacerbated the secondary apoptotic damage in the 7-day-old rat pups	NMDA
Huh et al. [63]	Rats	11 days	Controlled cortical impact	Typical axonal injury, neuroinflammation, and calpain activation were observed but seldom neuronal death	Microglia, astrocytes, *β*-APP
Kawamata et al. [66]	Rats	3–14 days	Repeated mild shaking brain injury	The postnatal day 7 rat pups had a significantly higher number of microhemorrhages than postnatal day 3 rat pups. No microhemorrhages were detected in the control rat pups and pups 5 weeks after shaking	Iron-positive reaction
Daniel et al. [72]	Rats	5 days	Low-intensity, high-duration rotating and high-intensity, low-duration movements	Hemorrhage was four times higher in the high-intensity group compared with low-intensity group	Reactive oxygen or nitrogen species, oxidative stress, and energy metabolism
Smith et al. [73]	Rats	6 days	Shaker	Shaking of the rat pups led to cortical hemorrhages, cortical tissue damage, and production of oxidative stress markers	Oxidative stress
Hanlon et al. [75]	Rats	11 days	Repeated injury neonatal model	Repeated injuries led to spatial learning and memory deficits and increased brain microglial and macrophage expression. Other typical damages to the brain included traumatic axonal injury, neuronal degeneration, and cortical and white matter atrophy	Microglia and macrophage
Friess et al. [84]	Piglets	3–5 days	Multiple impacts at either 1 day or 1 week apart	The multiple impacts led to a 43% mortality rate of the piglets, and the surviving animals displayed behavioral deficits and axonal injury Double rotation by one day apart led to a significantly higher mortality rate compared with a single rotation	*β*-APP
Raghupathi et al. [85]	Piglets	3–5 days	Rapid and inertial rotation of the head around the axial plane	Severe coma was observed in all piglets but recovered by 6 h Subdural and subarachnoid hemorrhage was evident in the frontal lobes	Hypoxia
Friess et al. [86]	Piglets	3–5 days	Mild level at 142 rad/s and moderate level at 188 rad/s	Behavioral deficits were observed in the moderate injury level during environmental exploration and visual-based problem solving Moderate injury level led to axonal injury	*β*-APP
Coats et al. [89,90]	Piglets	3–5 days old	Shaking frequency of 2–3 Hz, the average peak-to-peak angular velocity of 22.71 ± 3.49 rad/s, and average peak angular acceleration of 606.21 ± 160.30 rad/s^2^	Ocular hemorrhages were observed in 73% of the 51 piglets compared with a single head rotation The 30 s continuously rotated piglets had more hypoxic–ischemic injury	*β*-APP
Naim et al. [88]	Piglets	3–5 days female	Inertial loading device rapidly rotated the head of the animal through a 90 to 110° arc in the axial plane	Injuries at deep white matter of the frontal lobes, parietal and temporal lobes, or brainstem and axonal injuries measured by *β*-APP 6 days after injury were not affected by treatment	*β*-APP
Umstead et al. [91]	Piglets	6 months	Porcine ex vivo hypertensive experiment	A combined loading of shaking and hypertension contributes to retinal hemorrhaging in AHT	Hypertensive stress
Costine-Bartell et al. [94]	Piglets	1 week and 1 month	Cortical impact, seizure, et al. multiple insults	Infant piglets were clinically worse than their toddler counterparts	*β*-APP
Ibrahim et al. [96]	Piglets	4 weeks	Rotational accelerations and a single non-impact head rotation acceleration in the axial direction	A significant difference for both SAH scores and brain volumes of axonal injury when comparing the results of the 4-week-old piglets with 5-day-old counterparts; a larger percentage volume of *β*-APP staining for axonal injury was found in moderate-acceleration animals compared with shams and low-acceleration animals	*β*-APP, neurofilament (NF68)
Anderson et al. [98] Finnie et al. [99] Sandoz et al. [100]	Lambs	7–10 days	Manually subjected to vigorous and multiple-episode shaking for 30 min	Serial injuries, including BBB disruption, axonal damage, brainstem injury, and craniocervical junction damage were observed during the studies	*β*-APP
Finnie et al. [101]	Lambs	7–10 days	Subject to free shaking 10 × 30 s for 30 min	Caused extra-axial hemorrhages in all lambs Significant *β*-APP-positive neuronal perikaryons were observed in all injured lambs	*β*-APP, glial fibrillary acidic protein
Serbanescu et al. [104]	Cats	3–4 weeks	The kittens and rabbit were bitten on the haunches and posterior spine, followed by aggressive 4–6 side-to-side shakes	No retinal or optic nerve sheath hemorrhage was observed	Retinal hemorrhages

AHT abusive head trauma, *β*-APP *β*-amyloid precursor protein, NMDA *N*-methyl-D-aspartate receptor, SAH subarachnoid hemorrhage.

## Data Availability

The materials and resources in this study are available from the corresponding author upon reasonable request.

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
