# Peer review of "Abusive Head Trauma Animal Models: Focus on Biomarkers"

_ijms, 2023, doi:10.3390/ijms24054463_

Round 1
Reviewer 1 Report
Dear Authors,
When reading a review article on Abusive Head Trauma, the most critical points that must be scrutinized are the following. How AHT is defined in the cited article, what percentage of cases of AHT are confirmed in the group of patients studied, and whether cases of suspected AHT are included.
Since the diagnosis of AHT is by nature medically difficult, requires careful consideration of the patient's social background, and is completed through multidisciplinary collaboration, it must be said that while of great academic interest, it is unlikely that the biomarker will significantly influence the decision. Unfortunately, the accuracy of AHT diagnosis with certain markers, while ideal, should not be high enough to assure that it does not create false convictions in court.
At the very least, the table must state how many confirmed cases of AHT have been diagnosed, or the accusation of circular reasoning will be unavoidable.
The references cited in this article, which was compiled to provide a comprehensive review of AHT biomarkers, are all more than 10 years old, and some of the basic experimental systems do not reproduce the so-called "abusive pathology. The chapters introducing the results of basic research are too verbose, making it difficult to understand the main points, and there are serious problems in understanding whether this is a phenomenon unique to AHT or not, or whether this is also true of accidental head trauma.
In addition, the extremely excellent literature recently presented by Costine-Bartell et al. in Boston is not cited. Unfortunately, we believe that this literature does not provide new horizons for the diagnosis of AHT.
Author Response
We thank the reviewers for their time and insightful critiques that have enabled us to improve our manuscript. We have made every effort to address each point raised by the reviewers. Please see our responses to Reviewer 1’s comments below.
Reviewer 1:
Dear Authors,
When reading a review article on Abusive Head Trauma, the most critical points that must be scrutinized are the following. How AHT is defined in the cited article, what percentage of cases of AHT are confirmed in the group of patients studied, and whether cases of suspected AHT are included.
Since the diagnosis of AHT is by nature medically difficult, requires careful consideration of the patient's social background, and is completed through multidisciplinary collaboration, it must be said that while of great academic interest, it is unlikely that the biomarker will significantly influence the decision. Unfortunately, the accuracy of AHT diagnosis with certain markers, while ideal, should not be high enough to assure that it does not create false convictions in court.
Response: We appreciate the reviewer’s comments. We agree that the diagnosis of AHT is medically challenging. No single modality can preciously diagnose clinical AHT. We provided AHT cases and total enrolled cases in the revised Table 1 (page 5, line 228). In the AHT cases cited in the manuscript, all papers stated that the AHT cases included a witnessed or confessed episode of shaking or diagnosed by a child protection team. While the biomarkers are not specific to clinical AHT, they should help clinicians raise the awareness that the brain may be injured, which is challenging to assess by traditional physical exams or neuroimaging approaches.
At the very least, the table must state how many confirmed cases of AHT have been diagnosed, or the accusation of circular reasoning will be unavoidable.
Response: We provided the confirmed AHT cases and total enrolled cases in Table 1 (page 5, line 228).
The references cited in this article, which was compiled to provide a comprehensive review of AHT biomarkers, are all more than 10 years old, and some of the basic experimental systems do not reproduce the so-called "abusive pathology. The chapters introducing the results of basic research are too verbose, making it difficult to understand the main points, and there are serious problems in understanding whether this is a phenomenon unique to AHT or not, or whether this is also true of accidental head trauma.
Response: We updated the references (pages 6, 8, 10, 13) and added more recent AHT animal studies to the revised manuscript (highlighted in the reference section). We agree that the basic pre-clinical research on AHT models does not reproduce the pathology of child abuse. In addition, we removed redundant contents in the manuscript and added critical analysis to each animal model. Currently, no reliable animal models have been able to reproduce the complete range of neuropathologic changes in AHT. As we discussed in the manuscript, the biomarkers are not specific to AHT and cannot be used to diagnose AHT accurately. However, the biomarkers could be used to screen for evidence of brain injuries (page 18, line 736).
In addition, the extremely excellent literature recently presented by Costine-Bartell et al. in Boston is not cited. Unfortunately, we believe that this literature does not provide new horizons for the diagnosis of AHT.
Response: We provided the new reference presented by Costine-Bartell et al. in the revised manuscript (page 10, line 464). The multifactorial injury cascades animal model is helpful in studying AHT-induced post-traumatic disorders and biomarker changes at different developmental stages.
Reviewer 2 Report
The manuscript presented from Rahul M Nikam et al., entitled "Abusive head trauma animal models: focus on biomarkers" is interesting. However the authors presented this manuscript as a review concerning animal models, but reading the text they mention only mice and rat, no other model organisms as xenopus, zebrafish, etc. I would suggest the authors to change the title in mouse and rat animal models and abstract or add further text concerning other species.Author Response
We thank the reviewers for their time and insightful critiques that have enabled us to improve our manuscript. We have made every effort to address each point raised by the reviewers. Please see our responses to Reviewer 2’s comments below.
Reviewer 2:
The manuscript presented from Rahul M Nikam et al., entitled "Abusive head trauma animal models: focus on biomarkers" is interesting. However the authors presented this manuscript as a review concerning animal models, but reading the text they mention only mice and rat, no other model organisms as xenopus, zebrafish, etc. I would suggest the authors to change the title in mouse and rat animal models and abstract or add further text concerning other species.
Response: Most AHT animal models are developed in young rodents, piglets, and lambs. The current review mainly focuses on AHT models and biomarker findings in these species. We included AHT animal models developed in other species, including cats, rabbits, zebrafish larvae, and Xenopus laevis tadpoles (page 13, section 4.5).
Reviewer 3 Report
The authors have written a review about abuse head trauma in infants and children, focusing on molecular markers and animal models in use. The topic is important, given the incidence of abuse head trauma, and such a review can help focus further research. The manuscript is well written, as well.
Several changes and/or additions to the manuscript should improve its quality, as follows:
1. In INTRODUCTION, lines 68-69, the authors should be aware that MRI can be used to identify subarachnoid hemorrhage, as shown by Mitchell, et al: Journal of Neurology, Neurosurgery, and Psychiatry 70:205-211, 2001.
2. In Section 4.1, the authors should offer a critical analysis of which mouse model is best, and why.
3. In section 4.2, the authors should offer a critical analysis of which rat model is best, and why.
4. In section 4.3, the authors should offer a critical analysis of which piglet model is best, and why.
5. In section 4.4, the authors should offer a critical analysis of which lamb model is best, and why.
6. In DISCUSSION, the authors should offer a critical analysis of which model they reviewed is best overall, including the ability to incorporate marker analysis into the model.
7. The CONCLUSIONS should be shorter (3-4 sentences maximum).
Author Response
We thank the reviewers for their time and insightful critiques that have enabled us to improve our manuscript. We have made every effort to address each point raised by the reviewers. Please see our responses to Reviewer 3’s comments below.
Reviewer 3:
The authors have written a review about abuse head trauma in infants and children, focusing on molecular markers and animal models in use. The topic is important, given the incidence of abuse head trauma, and such a review can help focus further research. The manuscript is well written, as well.
Several changes and/or additions to the manuscript should improve its quality, as follows:
1. In INTRODUCTION, lines 68-69, the authors should be aware that MRI can be used to identify subarachnoid hemorrhage, as shown by Mitchell, et al: Journal of Neurology, Neurosurgery, and Psychiatry 70:205-211, 2001.
Response: We appreciate the reviewer’s comments. We revised the statement to define MRI role more appropriately in subarachnoid hemorrhage (page 2, line 75).
2. In Section 4.1, the authors should offer a critical analysis of which mouse model is best, and why.
Response: We added an analysis of the mouse model to section 4.1 (page 7, line 314).
3. In section 4.2, the authors should offer a critical analysis of which rat model is best, and why.
Response: We added an analysis of the rat model to section 4.2 (page 9, line 391).
4. In section 4.3, the authors should offer a critical analysis of which piglet model is best, and why.
Response: We added an analysis of the piglet model to section 4.3 (page 12, line 555).
5. In section 4.4, the authors should offer a critical analysis of which lamb model is best, and why.
Response: We added an analysis of the lamb model to section 4.4 (page 13, line 597).
6. In DISCUSSION, the authors should offer a critical analysis of which model they reviewed is best overall, including the ability to incorporate marker analysis into the model.
Response: We added a critical analysis, including AHT animal models and biomarker analysis, to the discussion section (page 18, line 727).
7. The CONCLUSIONS should be shorter (3-4 sentences maximum).
Response: We revised the conclusion section (page 18, line 746).
Round 2
Reviewer 1 Report
Dear Authors,
Revised paper has been peer-reviewed. Frankly, I was impressed by the sincerity with which you addressed my points in such a short period of time. I am confident that this revision will avoid the so-called circularity and that this review article will play a major role in the future elucidation of the pathogenesis of the disease. I do not know of a more comprehensive summary of the pathogenesis of AHT. I would like to acknowledge its receipt and confirm that it has been extremely valuable in this sense. Congratulations and best wishes for the future.
Reviewer 2 Report
The authors satisfied all my concerns.